

# Assessment of urinary oxidative stress biomarkers associated with fine particulate matter (PM2.5) exposure in Chiang Mai, Thailand

Shamsa Sabir[1], Surat Hongsibsong[1,2], Hataichanok Chuljerm[1,2], Wason Parklak[2], Sakaewan Ounjaijean[1,2], Puriwat Fakfum[2], Sobia Kausar[1] and Kanokwan Kulprachakarn[1,2]

[1] School of Health Sciences Research, Research Institute for Health Sciences, Chiang Mai University, Chiang Mai, Thailand
[2] Research Center for Non-infectious Diseases and Environmental Health, Research Institute for Health Sciences, Chiang Mai University, Chiang Mai, Thailand

## ABSTRACT

**Background.** Exposure to fine particulate matter (PM2.5) is known to increase oxidative stress, impacting health adversely. This study examines the relationship between PM2.5 exposure and oxidative stress biomarkers in Chiang Mai, Thailand.

**Methods.** A pilot prospective observational study was conducted in Samoeng District, Chiang Mai, including 25 healthy participants (age 25–60 years). Urine samples were collected during high (March–April 2023) and low (May–July 2023) PM2.5 seasons. PM2.5 concentrations were monitored daily from the Northern Thailand Air Quality Health Index (NTAQHI) system. Biomarkers analyzed included 1-hydroxypyrene (1-OHP) using high-performance liquid chromatography (HPLC), malondialdehyde (MDA) *via* Spectrophotometry, and 8-epi-prostaglandin F2$\alpha$ (8-epi-PGF2$\alpha$) with Enzyme-linked Immunosorbent Assay (ELISA). Statistical analysis was performed using IBM SPSS Statistics 22.0.

**Results.** Significant increases in urinary 1-OHP, MDA, and 8-epi-PGF2$\alpha$ were observed during the high PM2.5 season compared to the low season. The mean concentration of PM2.5 was 67 $\mu$g/m$^3$ during high pollution and 7 $\mu$g/m$^3$ during low pollution. Elevated levels of these biomarkers indicate increased oxidative stress associated with higher PM2.5 exposure.

**Conclusions.** This study highlights a significant association between elevated PM2.5 levels and increased oxidative stress biomarkers in Chiang Mai, Thailand. The findings suggest that exposure to higher concentrations of PM2.5 contributes to oxidative stress, potentially leading to adverse health outcomes.

Corresponding author
Kanokwan Kulprachakarn,
kanokwan.kul@cmu.ac.th

# INTRODUCTION

Air pollution, resulting from both human activities and natural sources such as vehicle and industrial emissions, agricultural residue burning, biomass burning, and forest fires,

has significant transboundary effects, especially in regions like Southeast Asia (*Thongtip et al., 2022*). A WHO study estimates that seven million people die annually as a result of air pollution, both indoors and outdoors (*Irfan, 2024*). In developing countries, rapid urbanization and industrialization have exacerbated this problem, leading to increased exposure to pollutants like PM2.5, which poses severe health risks (*Sukkhum et al., 2022*).

PM2.5, fine particulate matter or particulate matter with a diameter of less than 2.5 micrometers, is particularly concerning due to its ability to penetrate deep into the respiratory system and enter the bloodstream (*Amnuaylojaroen & Parasin, 2023*). Its small size and large surface area enable PM2.5 to carry various toxic substances, including polycyclic aromatic hydrocarbons (PAHs), which are known to induce oxidative stress and inflammation. Epidemiological and toxicological studies have linked PM2.5 exposure to a range of health issues, including cardiovascular diseases, respiratory problems, and various forms of cancer (*Bhatnagar, 2022*). Additionally, oxidative stress and DNA damage are critical mechanisms through which PM2.5 exerts its harmful effects, facilitated by reactive oxygen species (ROS) production (*Liu et al., 2024*).

The mechanism of oxidative stress illustrated in Fig. 1 induced by PM2.5 and PAHs is a critical pathway in understanding the health impacts of air pollution. As illustrated in Fig. 1, the small size of PM2.5 allows it to penetrate deep into the respiratory system and enter the bloodstream, where it can transport toxic substances such as PAHs. These PAHs are activated through cytochrome P450 enzymes, converting them into reactive intermediates like epoxides and quinones (*Boström et al., 2002*). These intermediates can covalently bind to DNA, forming DNA adducts that can lead to mutations and carcinogenesis. The process also generates reactive oxygen species (ROS), causing further cellular damage. This dual mechanism highlights PAHs' significant role in promoting carcinogenic processes. Understanding this mechanism is vital for identifying potential biomarkers, such as 1-hydroxypyrene (1-OHP), malondialdehyde (MDA), and 8-epi-prostaglandin F2 alpha (8-epi-PGF2$\alpha$), which can be used to monitor exposure and assess the biological impact of PM2.5 on human health.

PAHs have been shown to create ROS through the redox cycle, which can induce oxidative modification of DNA and lipids *in vivo* (*Fujitani et al., 2023*). Urinary 1-hydroxypyrene (1-OHP) is commonly accepted as a valid biomarker for measuring internal PAH exposure (*Mucha et al., 2006*). In this research of 3-year-old children, 1-OHP concentrations were effectively employed to evaluate ambient exposure to PAHs, suggesting its applicability for population-level biomonitoring.

Biomarkers like MDA and 8-iso-PGF2$\alpha$ have been widely used in studies assessing oxidative stress related to air pollution exposure (*Zhao et al., 2018*). MDA is a well-known biomarker for oxidative stress, and its elevated levels have been linked to exposure to pollutants like PM2.5 and $O_3$. Urinary MDA levels have been used as a valid biomarker for oxidative stress caused by air pollution exposure, notably in children, the elderly, and people with pre-existing respiratory diseases (*Kim et al., 2009*). Despite its potential, the application of MDA as a biomarker in epidemiological studies concerning air pollution remains relatively limited.

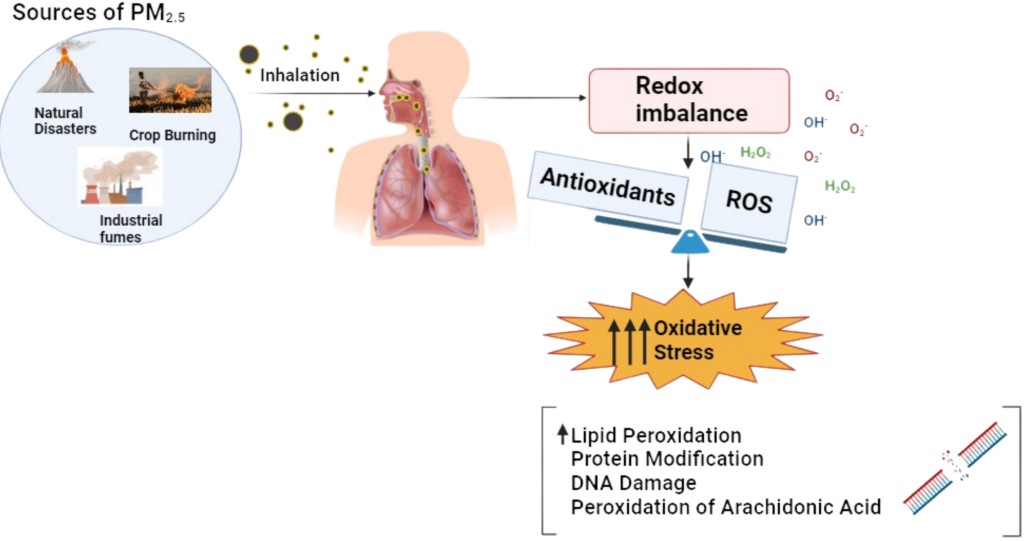

**Figure 1** Mechanism of oxidative stress induced by PM2.5 exposure, showing the imbalance between ROS and antioxidants leading to oxidative damage.

On the other hand, 8-epi-PGF2$\alpha$, part of the F2-isoprostane family, is a nonenzymatic product of arachidonic acid peroxidation and is recognized as a sensitive biomarker for oxidative status (*Il'yasova, Scarbrough & Spasojevic, 2012*). The metabolite 2,3-dinor-8-iso-prostaglandin F2$\alpha$ is particularly useful as a biomarker for the formation of 8-epi-PGF2$\alpha$ and lipid peroxidation *in vivo*, providing insights into the oxidative damage occurring within the body due to environmental and other stressors.

Chiang Mai, particularly the Samoeng district, experiences significant air pollution issues, primarily during the dry season due to agricultural residue burning and forest fires (*Jainonthee et al., 2022*). The region's topography, characterized by mountains and valleys, exacerbates air quality problems, trapping pollutants and leading to extended periods of poor air quality (*Supasri et al., 2023*). Despite these challenges, there is limited data on the health impacts of air pollution in this region, particularly concerning oxidative stress biomarkers.

Given the significant impact of PM2.5 on public health and the limited data available on oxidative stress biomarkers associated with air pollution in Chiang Mai, this study aims to investigate the relationship between PM2.5 exposure and oxidative stress biomarkers in the region. The primary objectives are to assess urinary levels of 1-OHP, MDA, and 8-epi-PGF2$\alpha$, evaluate their association with PM2.5 exposure, and explore the implications for public health in the Samoeng district. This research seeks to provide valuable insights into the biological effects of air pollution, contributing to developing effective strategies for mitigating its adverse health impacts in northern Thailand.

## MATERIALS AND METHODS

### Study area

The focus area was chosen to be Chiang Mai province (*Jarernwong, Gheewala & Sampattagul, 2023*). With a total area of 20,107 km$^2$, it is the largest province in the north of Thailand. The geographical location of the Chiang Mai province is shown in Fig. 2. It has 25 districts, with a total population of roughly 1.78 million based on the Department of Provincial Administration of Thailand in 2020.

A unique area of Chiang Mai province, Samoeng District, was selected for sample collection because of its unique environment, including higher levels of particulate matter PM2.5 during the burning season, as well as its rural and agricultural setting, making it a perfect location to study health impacts of pollution (*Paesrivarotai & Tanaksaranond, 2021*).

### Study population

This pilot, prospective observational study included 25 healthy participants from the Samoeng district of Chiang Mai province, Thailand, which was selected based on its relevance to the study's objectives of assessing environmental exposure. The inclusion criteria were individuals aged 25–60 years, no history of chronic respiratory or cardiovascular diseases, and a minimum residency of 5 years in the district to ensure homogeneity and relevance to environmental exposure assessment. Despite the small sample size, this pilot study aims to provide preliminary insights to guide larger-scale investigations. Exclusion criteria included any underlying diseases, recent operations, certain chronic conditions, pregnancy, drug abuse, psychological disorders, and infections.

### Data collection

Data collected from participants includes demographics (age, gender, smoking, alcohol drinking, underlying diseases, marital status, education, occupation, family income and financial support), physical examination (height, weight, BMI, waist circumference, hip circumference, diastolic BP, systolic BP, and heart rate). Urine samples were examined for biomarkers (1-OHP, MDA, and 8-epi-PGF2$\alpha$) detection.

Urine samples were collected during high (March–April 2023), and low (May–July 2023) PM2.5 seasons, which were provided by Asst. Prof. Kanokwan Kulprachakarn, Ph.D., from the research project entitled "Health Risk Assessment and Association between Metabolic and Hormonal Derangements in People Exposed to In-house or Ambient PM2.5-Bond Chemicals". The samples were stored at −20 °C until further analysis.

The concentrations of PM2.5 was measured and recorded daily, Ambient air quality data for PM2.5 was retrieved from a single monitoring location at Samoeng Hospital, located in the Samoeng district of Chiang Mai. The data was obtained from the Northern Thailand Air Quality Health Index (NTAQHI) system (https://www2.ntaqhi.info/), operated by the Research Institute for Health Sciences (RIHES), Chiang Mai University. PM2.5 measurements were collected using sensors developed by the Environmental and Occupational Health Unit at RIHES. Quality control procedures and data averaging

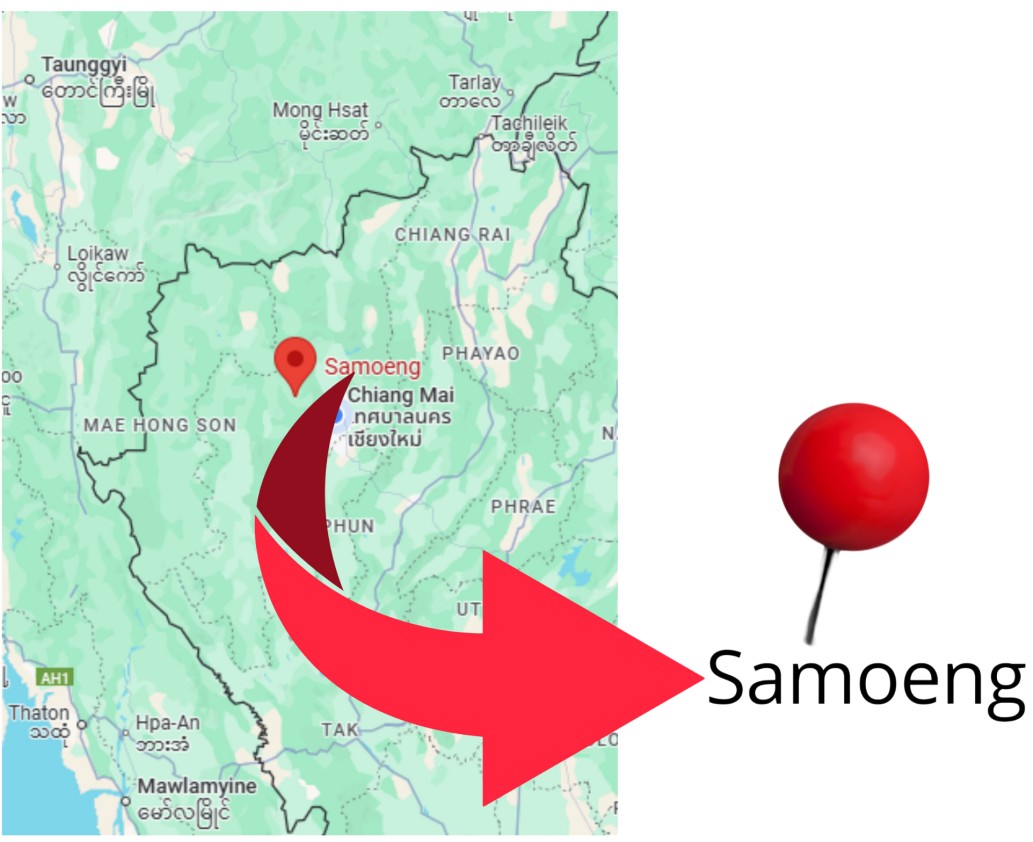

**Figure 2** **Map showing the study area in Chiang Mai, Thailand.**

techniques followed the protocols established by RIHES to ensure data accuracy and reliability.

## Biomarkers analysis

### Creatinine (Cr)

The amount of Cr in the samples was measured using the spectrophotometric Jaffé method, which relies on the reaction of Cr with picric acid in an alkaline pH solution (*Campos et al., 2011*). For this purpose a creatinine assay kit (Colorimetric) (ab204537) was used which is a complete kit for the quantitative determination of creatinine in urine.

### 1-Hydroxypyrene (1-OHP)

The urine samples, collected in March 2023 and stored for one year, were used for 1-OHP analysis with modifications by the method of *Sutan, Naksen & Prapamontol (2017)*. To a 2.5 mL urine sample, 1M HCl was added to adjust the pH to 5.0, followed by 2.5 mL of 0.1M acetate buffer and 6.25 μL of $\beta$-glucuronidase from *Helix pomatia*. The mixture was vortexed and incubated at 37 °C for 2 h. After incubation, the samples were processed using a Vertipak C18 three mL solid-phase extraction (SPE) cartridge, pre-conditioned with methanol and water. The samples were loaded onto the cartridge, washed with water and 20% methanol, and then eluted with methanol. The eluted solution was filtered,

evaporated, and reconstituted in 100 μL methanol for HPLC analysis. A 20-μL injection was performed using an Agilent 1260 Infinity HPLC system with a 45% water and 55% acetonitrile mobile phase at 0.80 mL/min. Separation was achieved using an InfinityLab Poroshell 120 EC-C18 column at 25 °C, and detection was performed using fluorescence at 242 nm (excitation) and 388 nm (emission) for 20 min. A standard curve (0.00125–2.50 ng/mL) was used for quantification, with limits of detection and quantification of 0.2634 and 0.7984 ng/mL, respectively.

### Thio barbituric acid-reacting substances (TBARS)

TBARS were measured by the method of *Campos et al. (2011)*. In summary, 140 μL of urine was mixed in a vortex with 33 μL of 0.01% BHT (in absolute ethanol), one mL of 1% phosphoric acid, and 300 μL of 42 mmol/L TBA (dissolved in water and heated). Following a 45-minute incubation period in boiling water, 1.4 mL of 1-butanol was added to each tube and the tubes were allowed to cool on ice. Using a UVmini-1240 Shimadzu spectrophotometer (Shimadzu, Tokyo, Japan), the absorbance of the supernatant was measured at 535 nm after a 15-minute centrifugation ($2000 \times g$). To create the standard absorption curve, MDA was dissolved in 20 mmol/L of phosphate buffer (pH 7.0).

### 8-epi-prostaglandin F2 alpha (8-epi-PGF2$\alpha$)

Urinary 8-epi-PGF2$\alpha$ was determined by the commercial ELISA kit (Elabscience) according to the manufacturer's instruction. The ELISA kit uses the Competitive-ELISA principle, using a pre-coated micro plate with 8-epi-PGF2$\alpha$. The enzyme competes with a fixed amount on the solid phase supporter for specific sites. Excess conjugate and unbound sample are washed away, Avidin-Horseradish Peroxidase conjugate is added, and TMB substrate solution is added. Optical density (OD) is measured.

## Ethical considerations

The study protocol was approved by Human Experimentation Committee Research Institute for Health Sciences (RIHES), Chiang Mai University, Chiang Mai, Thailand on 19 January 2023 (Project No. 03/2023). Written informed consent was obtained from all subjects involved in the study.

## Statistical analysis

The data was analyzed using IBM SPSS Statistics 22.0. To assess the normality of the data, the Shapiro–Wilk test was performed. For variables that were normally distributed, the results are presented as mean $\pm$ standard deviation (SD), whereas for non-normally distributed variables, the results are reported as median (interquartile range (IQR)). To compare biomarker concentrations across the high and low PM2.5 seasons, the Wilcoxon signed-rank test was used. A *p*-value of $<0.05$ was considered statistically significant. To further assess the relationship between the predictor variable (PM2.5) and the outcome variables (1-hydroxypyrene (1-OHP), malondialdehyde (MDA), and 8-epi-prostaglandin F2$\alpha$ (8-epi-PGF2$\alpha$)), a mixed-effects generalized linear model (GLM) was employed. The GLM utilized a Gaussian family with an identity link function, and fixed effects for predictors were estimated while accounting for clustering by participant code as a random

intercept. Robust standard errors were applied to address within-group correlation and heteroscedasticity. The results are reported for fixed effects only, as random effect estimates were not central to the research objectives.

## RESULTS

### Baseline characteristics of participants

This study assessed 25 healthy participants from the Samoeng district in Chiang Mai province, Thailand. The participants had a mean age of 48.1 years (SD = 14.6), with a gender distribution of 60% female and 40% male. Most participants were non-smokers (84%), and 76% reported alcohol consumption. Educational backgrounds varied, with 60% having less than high school education and 40% having more than high school education. The occupational breakdown showed that 52% of participants worked in agriculture, 8% in service, and 40% in other sectors. Table 1 shows the baseline characteristics of the participants in terms of mean and standard deviation for continues variables and frequency and percentage for categorical variables.

### PM2.5 concentration during high and low pollution seasons

The daily PM2.5 concentration data were obtained from the NTAQHI website, managed by RIHES CMU, for the periods of March–April and May–July 2023. The mean concentrations of PM2.5 during both high and low pollution seasons are presented in Fig. 3. During the high pollution season (March–April 2023), PM2.5 levels were significantly elevated, with the highest concentration observed in April (88.3 $\mu g/m^3$), followed by March (67.3 $\mu g/m^3$). In contrast, the low pollution season (May–July 2023) exhibited substantially lower PM2.5 concentrations, with mean levels of 15.8 $\mu g/m^3$ in May and 4.1 $\mu g/m^3$ in July. The variation between these months highlights the distinct seasonal air quality differences in the Samoeng District of Chiang Mai.

This variation in PM2.5 levels between the two seasons was critical for analyzing the seasonal impact of air pollution on urinary oxidative stress biomarkers in participants. The average PM2.5 concentration for the high pollution season was calculated at 67 $\mu g/m^3$, while the low pollution season averaged seven $\mu g/m^3$. These differences provided a strong basis for assessing the correlation between PM2.5 exposure and biomarker levels, particularly for oxidative stress markers such as 1-OHP, MDA, and 8-epi-PGF2$\alpha$.

### Urinary biomarkers and PM2.5 concentration by season

In this study, the authors observed significant differences in the urinary concentrations of oxidative stress biomarkers between high and low PM2.5 seasons, which are depicted in Fig. 4. The graph presents the median concentrations of 8-epi-PGF2$\alpha$, 1-OHP, and MDA across the two seasons, highlighting the impact of seasonal PM2.5 exposure on these biomarkers.

The concentration of 8-epi-PGF2$\alpha$, measured in pg/mg creatinine, was significantly higher during the high PM2.5 season (median = 139.43 pg/mg, Q1 = 80.63, Q3 = 187.54) compared to the low PM2.5 season (median = 54.22 pg/mg, Q1 = 26.74, Q3 = 122.76), with a $p$-value of 0.016. This indicates an elevated oxidative stress level during periods of higher air pollution.

Table 1 Baseline characteristics of the participants (N = 25).

| Characteristic | N (%) |
|---|---|
| Age (years) | |
| Mean ±SD | 48.1 ± 14.6 |
| Gender | |
| Male | 10 (40.0%) |
| Female | 15 (60.0%) |
| Smoking | |
| No | 21 (84.0%) |
| Yes (included smoking and quit smoke) | 4 (16.0%) |
| Number of smoke/day | |
| Mean ± SD (min–max) | 2 ± 1.7 (1–5) |
| History of quitting smoking | |
| Quit smoking (years) | |
| Mean ± SD (min–max) | 18.3 ± 2.9 (15–20) |
| Duration of smoking (years) | |
| Mean ± SD (min–max) | 21.7 ± 14.4 (5–30) |
| Alcohol consumption | |
| No | 6 (24.0%) |
| Yes | 19 (76.0%) |
| Education Level | |
| Less than high school | 15 (60.0%) |
| More than high school | 10 (40.0%) |
| Occupation | |
| Agriculture | 13 (52.0%) |
| Service | 2 (8.0%) |
| Others | 10 (40.0%) |

The concentration of malondialdehyde (MDA), measured in $\mu$M/mg creatinine, was also significantly higher during the high PM2.5 season (median = 3.15 $\mu$M/mg, Q1 = 2.83, Q3 = 4.11) compared to the low PM2.5 season (median = 2.45 $\mu$M/mg, Q1 = 1.96, Q3 = 3.05), with a p-value of 0.006. The elevated MDA levels during high PM2.5 exposure further support the hypothesis that increased air pollution contributes to oxidative stress.

Similarly, the urinary concentration of 1-hydroxypyrene (1-OHP), expressed in mg/g creatinine, showed a significant increase during the high PM2.5 season (median = 0.09 mg/g, Q1 = 0.06, Q3 = 0.17) compared to the low PM2.5 season (median = 0.04 mg/g, Q1 = 0.02, Q3 = 0.11), with a p-value of 0.001. The increased levels of 1-OHP suggest higher internal exposure to polycyclic aromatic hydrocarbons (PAHs) during periods of elevated PM2.5 levels.

The Wilcoxon signed-rank test was applied in Table 2 to assess these differences, and all p-values were found to be significant, indicating a robust association between increased PM2.5 exposure and elevated urinary biomarkers of oxidative stress. These findings underscore the heightened oxidative stress during periods of high air pollution,

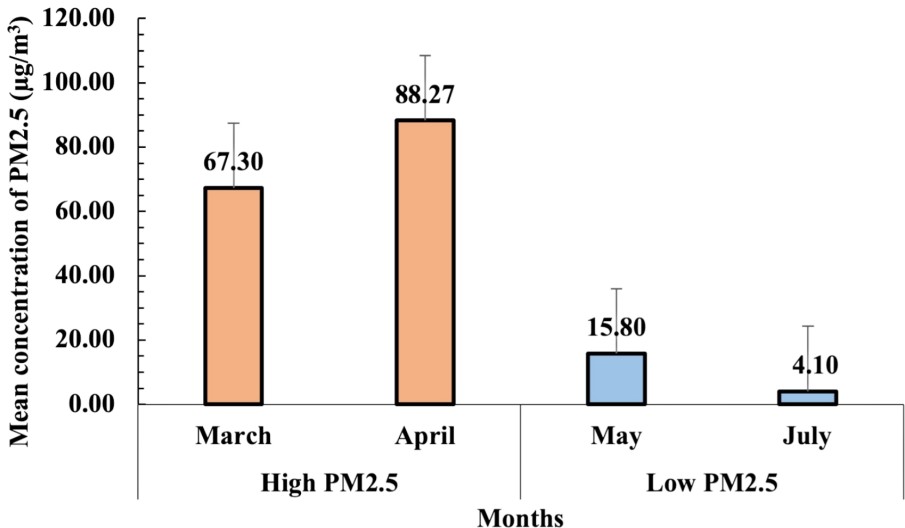

**Figure 3** **Mean concentrations of PM2.5 ($\mu$g/m³) recorded during high (March–April 2023) and low (May–July 2023) pollution seasons.** The error bars represent the standard error of the mean (SEM).

particularly in the Samoeng District, where residents, predominantly farmers, are more susceptible to PM2.5 exposure due to agricultural practices like stubble burning.

This analysis provides clear evidence of the health risks associated with seasonal variations in PM2.5 levels, reinforcing the need for targeted interventions to reduce exposure, particularly in vulnerable populations. The significant changes in these biomarkers reflect the biological impact of air pollution and suggest potential pathways through which PM2.5 exposure may lead to adverse health outcomes.

## Biomarkers in relation to PM2.5 exposure, age, gender, and smoking status

The mixed-effects REML regression model was used in Table 3 to analyze the associations between PM2.5 exposure and urinary oxidative stress biomarkers (1-OHP, MDA, and 8-epi-PGF2$\alpha$), adjusting for age, gender, and smoking status. Both adjusted and unadjusted models were presented to assess the influence of these covariates on the main outcomes.

For **1-hydroxypyrene (1-OHP)**, PM2.5 exposure was significantly associated with increased levels in both the adjusted ($\beta = 0.0036$, 95% CI [0.0001–0.007], $p = 0.043$) and unadjusted models ($\beta = 0.0035$, 95% CI [0.0001–0.007], $p = 0.044$). Other covariates, including age, gender, and smoking status, were not significant predictors in the adjusted model.

For **malondialdehyde (MDA)**, PM2.5 exposure exhibited a marginal association with increased levels in both the adjusted ($\beta = 0.038$, 95% CI [−0.001–0.077], $p = 0.056$) and unadjusted models ($\beta = 0.037$, 95% CI [−0.001–0.076], $p = 0.055$). Gender showed borderline significance in the adjusted model ($p = 0.066$), while age and smoking status were not significant predictors.

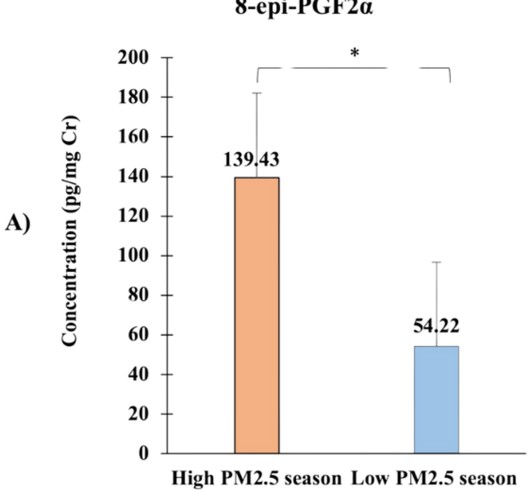

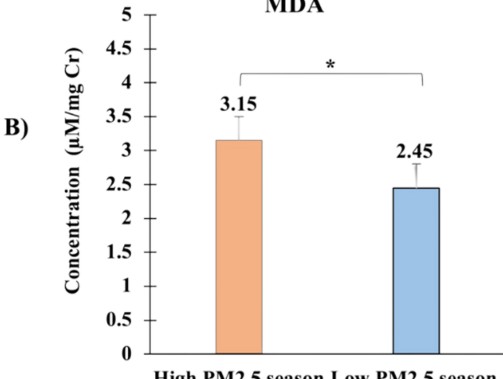

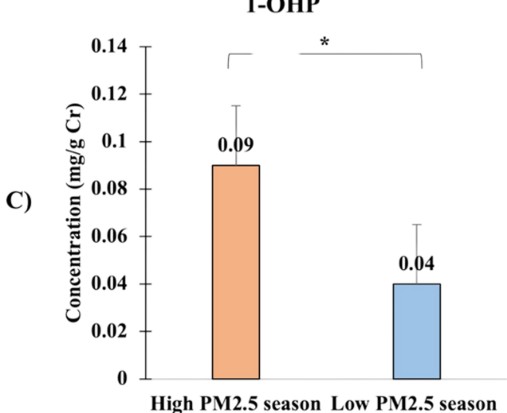

**Figure 4 Comparison of urinary concentrations of oxidative stress biomarkers, including (A) 8-epi-PGF2α, (B) Malondialdehyde (MDA), and (C) 1-hydroxypyrene (1-OHP) between high PM2.5 and low PM2.5 seasons.** Data are presented as median values with interquartile ranges (Q1, Q3) for each biomarker. The Wilcoxon signed-rank test was used to assess the significance of differences between seasons, with all *$p$-values $< 0.05$ indicating statistically significant increases in biomarker levels during the high PM2.5 season.

**Table 2  Comparison of urinary biomarkers between low and high PM2.5 levels using Wilcoxon signed-rank test.**

| Biomarker | Low PM2.5 (n = 25) | High PM2.5 (n = 25) | Median Diff. | p-value[**] |
|---|---|---|---|---|
| | Median (IQR) | Median (IQR) | | |
| 1-OHP (mg/g creatinine) | 0.04 (0.02, 0.06) | 0.09 (0.06, 0.15) | 0.04 (0.01,0.09) | 0.0008[*] |
| MDA (μM/mg creatinine) | 2.43 (1.95, 3.05) | 3.15 (2.83, 4.11) | 0.66 (0.09,1.77) | 0.0063[*] |
| 8-epi-PGF2α (pg/mg creatinine) | 54.54 (26.74, 122.76) | 130.52 (115.33, 223.75) | 72.22 (−6.59,122.69) | 0.0068[*] |
| PM2.5 Concentration (μg/m$^3$) | 4.17 (4.08, 4.25) | 41.64 (40.23, 44.60) | 37.56 (36.06,40.18) | 0.0075[*] |

Notes.

[*]P-values < 0.05 considered significant.

[**]Wilcoxon signed-rank test.

IQR, Interquartile Range.

For **8-epi-prostaglandin F2α (8-epi-PGF2α)**, PM2.5 exposure showed a strong and significant association with increased levels in both the adjusted ($\beta = 2.80$, 95% CI [0.966–4.64], $p = 0.003$) and unadjusted models ($\beta = 2.744$, 95% CI [0.901–4.587], $p = 0.003$). However, smoking status was significantly associated with a decrease in 8-epi-PGF2α levels ($\beta = -45.37$, 95% CI [−81.1 to −9.6], $p = 0.013$).

This unexpected association may be attributed to a reduced oxidative stress burden in smokers due to adaptive physiological responses. This finding highlights a potential biological complexity that warrants further exploration but does not alter the main outcome of the study.

Overall, the adjusted and unadjusted models yielded comparable results for PM2.5 exposure, indicating that smoking status and other covariates did not confound the main findings. The results clearly demonstrate the significant impact of PM2.5 on oxidative stress biomarkers, independent of demographic and behavioral factors.

## DISCUSSION

This pilot study provides novel insights into the relationship between PM2.5 exposure and oxidative stress biomarkers in Chiang Mai, Thailand. We sampled the same group of participants in both seasons, and the results showed higher biomarkers during the high PM2.5 season, indicating that PM2.5 exposure increased oxidative stress, even among smokers. Additionally, the combined exposure to both tobacco smoke and PM2.5 may have led to synergistic effects, where the cumulative impact of PM2.5 intensified oxidative stress, supporting the conclusion that PM2.5, rather than tobacco smoke alone, contributed to the increase in biomarkers. The observed increase in 1-OHP and 8-epi-PGF2α levels during periods of high PM2.5 exposure corroborates previous studies linking air pollution to oxidative stress. PM2.5 carries various toxic substances, including PAHs, which are metabolized into 1-OHP, a reliable biomarker for PAH exposure (*Luo, Stepanov & Hecht, 2019*; *Liu et al., 2023*). Similarly, the significant elevation in 8-epi-PGF2α, a marker of lipid peroxidation, aligns with research highlighting the role of air pollution in inducing oxidative damage and inflammation (*Glencross et al., 2020*; *Leni, Künzi & Geiser, 2020*).

The correlation between PM2.5 levels and MDA, though less pronounced than for 1-OHP and 8-epi-PGF2α, still underscores the oxidative stress induced by particulate
**Table 3  Fixed effects estimates from the mixed-effects REML regression model for the association between PM2.5 and elevated levels of urinary oxidative stress biomarkers.**

| Dependent Variable | Variables | Adjusted by age, gender, and smoke | | | | Unadjusted | | | |
|---|---|---|---|---|---|---|---|---|---|
| | | Coefficient ($\beta$) | SE | 95% CI | $p$-value | Coefficient ($\beta$) | SE | 95% CI | $p$-value |
| **1-OHP** | PM2.5 | 0.0036 | 0.002 | 0.0001, 0.007 | 0.043[*] | 0.0035 | 0.002 | 0.0001, 0.007 | 0.044[*] |
| | Age | 0.0037 | 0.003 | −0.0025, 0.009 | 0.243 | | | | |
| | Gender | 0.128 | 0.114 | −0.096, 0.351 | 0.262 | | | | |
| | Smoke | −0.076 | 0.06 | −0.194, 0.04 | 0.203 | | | | |
| **MDA** | PM2.5 | 0.038 | 0.019 | −0.001, 0.077 | 0.056 | 0.037 | 0.019 | −0.001, 0.076 | 0.055 |
| | Age | 0.034 | 0.022 | −0.009, 0.077 | 0.122 | | | | |
| | Gender | 1.165 | 0.632 | −0.075, 2.41 | 0.066 | | | | |
| | Smoke | −0.483 | 0.371 | −1.21, 0.24 | 0.193 | | | | |
| **8-epi-PGF2$\alpha$** | PM2.5 | 2.80 | 0.94 | 0.966, 4.64 | 0.003[*] | 2.744 | 0.94 | 0.901, 4.587 | 0.003[*] |
| | Age | −0.83 | 2.02 | −4.78, 3.12 | 0.681 | | | | |
| | Gender | −1.77 | 34.4 | −69.2, 65.7 | 0.959 | | | | |
| | Smoke | −45.37 | 18.2 | −81.1, −9.6 | 0.013 | | | | |

**Notes.**

1-OHP, 1-hydroxypyrene; MDA, Malondialdehyde; 8-epi-PGF2$\alpha$, 8-epi-prostaglandin F2$\alpha$.

*P-values < 0.05 indicating statistically significant association.

matter exposure. MDA, a byproduct of lipid peroxidation, has been widely used as a biomarker for oxidative stress in various studies, further validating our findings (*Cui et al., 2018*; *Zhang et al., 2023*).

Elevated urinary levels of specific isomers, such as iPF2$\alpha$-III and iPF2$\alpha$-VI, have been found in individuals with various health conditions, including cardiovascular disease, Alzheimer's disease, type 2 diabetes, Down syndrome, lung disorders, and heavy smokers (*Zhang et al., 2010*). The impact of PM2.5 on oxidative stress biomarkers, as evidenced by our study, suggests potential health risks, including cardiovascular and respiratory diseases, which have been extensively documented in the literature (*Brook et al., 2010*; *Schraufnagel et al., 2019*).

Our results also reveal a significant association between age and MDA levels, with older individuals exhibiting higher levels of oxidative stress. This finding is consistent with previous studies showing age-related susceptibility to oxidative damage due to the cumulative effects of environmental exposures over time (*Weary, 2023*) . Gender differences were observed, with females showing higher MDA levels, a finding that resonates with research suggesting that hormonal differences, particularly the presence of estrogen, may influence oxidative stress responses (*Viña et al., 2005*; *Berry, 2022*).

The observed inverse association between smoking status and 8-epi-prostaglandin F2$\alpha$ (8-epi-PGF2$\alpha$) levels is unexpected, as smoking is typically linked to increased oxidative stress. This counterintuitive finding may be attributed to adaptive physiological responses in chronic smokers. Prolonged exposure to cigarette smoke has been shown to upregulate antioxidant defenses, potentially mitigating oxidative damage (*Van der Vaart et al., 2004*).

Overall, our findings are consistent with a growing body of evidence linking air pollution, particularly PM2.5, to oxidative stress and the development of non-communicable diseases (NCDs). Oxidative stress is a key mechanism through which PM2.5 exposure contributes to the pathogenesis of cardiovascular and respiratory diseases, as well as cancer (*Wang et al., 2019*). The specific biomarkers analyzed in our study—1-OHP, MDA, and 8-epi-PGF2α— are widely recognized as indicators of oxidative stress, providing a robust framework for assessing the biological impact of air pollution (*Zhao et al., 2023*).

The findings of this study have important public health implications, particularly in regions like Chiang Mai, where seasonal air pollution is a significant concern. The association between PM2.5 exposure and oxidative stress biomarkers suggests that residents in highly polluted areas are at an increased risk of developing oxidative stress-related diseases. This underscores the need for targeted interventions to reduce air pollution exposure, particularly during the burning season. Additionally, the use of oxidative stress biomarkers in epidemiological studies could provide valuable insights into the long-term health effects of air pollution and inform public health policies aimed at mitigating these risks.

The strengths of this study lie in its focused, area-specific approach, targeting the Samoeng District, a region previously unexplored in this context. This district, primarily inhabited by farmers, presents a unique population with heightened exposure to PM2.5, particularly due to stubble burning, making the findings highly relevant to local environmental health concerns. The study's design, with repeated measurements during different PM2.5 seasons, allows for the observation of seasonal variations in oxidative stress biomarkers, providing a comprehensive view of the impact of air pollution on health. Additionally, the use of multiple biomarkers—8-epi-PGF2α, MDA, and 1-OHP— strengthens the evidence of oxidative stress due to PM2.5 exposure, and the significant correlations observed add to the robustness of the study's findings.

However, this study has some limitations. One of the primary limitations is the small sample size, which limits the generalizability of the findings. As a pilot study, the results should be interpreted with caution, and larger studies are needed to validate these findings. Furthermore, the study did not account for other potential confounding factors, such as diet, physical activity, and socioeconomic status, which could influence oxidative stress levels. The cross-sectional design of the study also limits the ability to draw causal inferences between PM2.5 exposure and oxidative stress biomarkers.

Future research should focus on expanding the sample size and incorporating longitudinal data to better understand the temporal relationship between PM2.5 exposure and oxidative stress. Additionally, exploring the role of other potential confounders, such as genetic susceptibility and lifestyle factors, could provide a more comprehensive understanding of the factors influencing oxidative stress. Investigating the combined effects of air pollution and other environmental exposures, such as indoor air pollution and chemical contaminants, could also shed light on the cumulative impact of multiple stressors on oxidative stress and health outcomes.

## CONCLUSION

In conclusion, this study highlights the significant impact of PM2.5 exposure on oxidative stress biomarkers, with higher levels observed during the high PM2.5 season in the Samoeng District. The strong associations found between PM2.5 and biomarkers such as 8-epi-PGF2$\alpha$, 1-OHP, and MDA reinforce the health risks posed by air pollution, particularly in rural areas prone to stubble burning. These findings underscore the urgent need for targeted strategies to mitigate air pollution and protect public health in vulnerable communities.

## ACKNOWLEDGEMENTS

The authors would especially like to thank the staff of Subdistrict Health Promoting Hospitals in Samoeng District, Chiang Mai, Thailand and also would like to thank Ms. Suthathip Wongsrithep for statistical assistance. Additionally, the authors would like to express sincere gratitude to Mr. Sharjeel Shakeel for his invaluable assistance in graph creation and manuscript review.

### Funding

This research project was supported by Fundamental Fund 2024, Chiang Mai University. This work was supported by Research Institute for Health Sciences, Chiang Mai University, Chiang Mai, Thailand, grant number 020/2567 and the Presidential Scholarship granted by Chiang Mai University, Chiang Mai, Thailand, grant number 8393(25)/1688. There was no additional external funding received for this study. The funders had no role in study design, data collection and analysis, decision to publish, or preparation of the manuscript.

### Grant Disclosures

The following grant information was disclosed by the authors:
Fundamental Fund 2024, Chiang Mai University.
Research Institute for Health Sciences, Chiang Mai University: 020/2567.
Presidential Scholarship granted by Chiang Mai University: 8393(25)/1688.

### Competing Interests

The authors declare there are no competing interests.

### Author Contributions

- Shamsa Sabir conceived and designed the experiments, performed the experiments, analyzed the data, prepared figures and/or tables, authored or reviewed drafts of the article, funding acquisition, and approved the final draft.
- Surat Hongsibsong analyzed the data, authored or reviewed drafts of the article, and approved the final draft.
- Hataichanok Chuljerm analyzed the data, authored or reviewed drafts of the article, and approved the final draft.

- Wason Parklak performed the experiments, authored or reviewed drafts of the article, and approved the final draft.
- Sakaewan Ounjaijean conceived and designed the experiments, authored or reviewed drafts of the article, and approved the final draft.
- Puriwat Fakfum performed the experiments, prepared figures and/or tables, and approved the final draft.
- Sobia Kausar analyzed the data, prepared figures and/or tables, and approved the final draft.
- Kanokwan Kulprachakarn conceived and designed the experiments, performed the experiments, analyzed the data, authored or reviewed drafts of the article, funding acquisition, and approved the final draft.

## Human Ethics

The following information was supplied relating to ethical approvals (i.e., approving body and any reference numbers):

Human Experimentation Committee Research Institute for Health Sciences (RIHES), Chiang Mai University, Chiang Mai, Thailand.

## Data Availability

The raw data and raw measurements are available in the Supplementary File.

## Supplemental Information

Supplemental information for this article can be found online at http://dx.doi.org/10.7717/peerj.19047#supplemental-information.

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
