# Peer review of "Assessment of urinary oxidative stress biomarkers associated with fine particulate matter (PM2.5) exposure in Chiang Mai, Thailand"

_PeerJ, doi:10.7717/peerj.19047_

## Round 0.1 · original submission · Major Revisions

Please revise the manuscript by following the reviewers' comments. A point-by-point response is needed when re-submitting your manuscript for re-consideration.

·

Basic reporting

This pilot study utilized the data and/or biospecimen from a larger cohort study to assess the effect of seasonal variation in PM2.5 on oxidative stress markers. While the oxidative stress pathway is accepted widely as the general mechanism for PM2.5 induced toxic effects, this study only adds to the evidence that is largely well accepted.

Some strengths include – first, a well demarcated and a significant seasonal variation in PM2.5 levels in the Samoeng district allows to test the studies research question relating to the relationship between ambient PM2.5 exposure, 1-OHP as an indicator of internal exposure to PAHs and oxidative stress biomarkers. Second, measured levels of 1-OHP, MDA and 8epiPGF2α. Multivariable regression analysis showing a significant association after adjusting for some confounders.

A major limitation of the study is the small sample size. Additionally, there is no clarity in delineating the effects of smoking from PM2.5 exposures on oxidative stress biomarkers. Since this study is part of a larger study, additional information on smoking history, duration, frequency should be used to properly account for its role on oxidative stress markers. Also, please check the references throughout and for specific comments in the pdf document.

Experimental design

please see my comments.

Validity of the findings

please see my comments.

Additional comments

Assessment of urinary oxidative stress biomarkers associated with fine particulate matter (PM2.5)
exposure in Chiang Mai, Thailand (#106470)

Line Comment
50 -51 The cited reference is not listed in the reference section. Moreover, suggest authors to use the most recent GBD estimates with correct reference
71 General mechanism of PAH induced ROS generation requires details, correctness and clarity. Please refer to PAH bioactivation to epoxides/quinones and ROS generation mechanism in Bostrom et al 2002, doi: 10.1289/ehp.110-1241197.
91-91 Author’s should cite papers focusing on environmental exposures of PAHs and use of 1-hydroxypyrene to monitor internal exposure to PAHs at population level.
100-101 Support your statement by citing relevant literature
147-149 It’s a small sample size! Please list your inclusion criteria and how it relates to the study’s objective
161-164 Need more details about the ambient air quality monitoring – data was retrieved from how many locations, type of air quality monitor, quality control and averaging technique.
172 Stored for how long?
300 Why GEE model and not GLM? With a small sample size and one time measurement, how is GEE is relevant?
Table 2 I have my concern with the regression output. Smoking is known to increase all the biomarkers included in this study. With over 80% of the participants (i.e., 21/25) reported as smokers, it should have some impact on the dependent variables. I suggest authors provide regression outputs for two seasons separately.
370-375 I disagree with the statement of ‘mixed results’ concerning co-exposures to tobacco smoke and PM pollution. There is a plethora of literature highlighting interactive effects of tobacco smoke, passive smoke and PM pollution. Tobacco smoke, especially among study subjects who are 40 plus years, and maybe smoking for over 5 years, would be a significant driver of oxidative stress markers. Moreover, PAHs are a major class of toxics from tobacco smoke. If the primary study has more information on tobacco smoke duration, urinary cotinine levels etc., this data should be used to delineate the effects of PM2.5-PAHs from the self-reported tobacco smoking.

More importantly, the listed references do not support mixed results as claimed by the authors.
Summary This pilot study utilized the data and/or biospecimen from a larger cohort study to assess the effect of seasonal variation in PM2.5 on oxidative stress markers. While the oxidative stress pathway is accepted widely as the general mechanism for PM2.5 induced toxic effects, this study only adds to the evidence that is largely well accepted.

Some strengths include – first, a well demarcated and a significant seasonal variation in PM2.5 levels in the Samoeng district allows to test the studies research question relating to the relationship between ambient PM2.5 exposure, 1-OHP as an indicator of internal exposure to PAHs and oxidative stress biomarkers. Second, measured levels of 1-OHP, MDA and 8epiPGF2α. Multivariable regression analysis showing a significant association after adjusting for some confounders.

A major limitation of the study is the small sample size. Additionally, there is no clarity in delineating the effects of smoking from PM2.5 exposures on oxidative stress biomarkers. Since this study is part of a larger study, additional information on smoking history, duration, frequency should be used to properly account for its role on oxidative stress markers. Also, please check the references throughout and for specific comments in the pdf document.

·

Basic reporting

Although the writing is clear with good English, the authors have not used appropriate analytical methods. This is a repeated measures design, with urine samples collected over two seasons in the same people.
The analysis appears to be a simple correlation between PM2.5 and biomarker levels in each season as if they had two independent samples supplemented by a regression analysis that does not take into account the repeated measures design. The correct way of analyzing these data are to note the dates that the urine samples were collected for each person; construct a database of daily PM2.5 data starting on the day of collection, to 30 days before (or at least a week before) in each season. This will allow the construction of various exposure moving averages of exposure in order to assess the exposure window associated with each biomarker level using a repeated measures analytic regression approach.

The overall purpose and scope of the paper is good, and the concepts are timely.

Experimental design

note my comment above. In the conduct of regression analyses--the distribution of the biomarkers should be examined as they may be skewed and require log transformation.

Validity of the findings

The data analysis is not correct.

Additional comments

The introduction can be substantially shorted - the literature review discussing oxidative stress belongs in the discussion. Most of the urine analytic methods can be put in a data supplement.

Reviewer 3 ·

Basic reporting

There are some problems with the raw data.

In the case of ID 2, the two creatinine values from visit 2 on the biomarker data sheet are different (creatinine (1.601 mg/ml) vs creatinine (0.002 g/ml)). These different creatinine values resulted in different results between MDA (2.739 µM/mg creatinine) and MDA (2192.308 µM/g creatinine) and between 8-epi-PGF2α (285.296 pg/mg creatinine) and 8-epi-PGF2α (228335.708 pg/g creatinine).

In the visit 1 data on the biomarker data sheet, all values for 8-epi-PGF2α (pg/mg creatinine) and 8-epi-PGF2α (pg/g creatinine) of all cases are odd. These two values should be calculated from 8-epi-PGF2α (pg/ml) divided by creatinine (mg/ml) or 8-epi-PGF2α (pg/ml) divided by creatinine (g/ml) as done in the visit 2 data. However, the values of the data are fixed numbers as measured values and two values are different and also different from calculated values with the provided data.

In Table 2, revise Smoking Status (smoker vs Non-Smoker) from MDA raw to “Smoking Status (smoker)”.

Experimental design

no comment

Validity of the findings

no comment

---

## Round 0.2 · accepted · Accept

The authors have addressed the reviewers' comments and clearly stated the limitations of this pilot study (small sample size, caution for generalization, etc). I am satisfied with the responses and endorse the publication of the study.

·

Basic reporting

No comment

Experimental design

No comment

Validity of the findings

No comment

Additional comments

Thank you for addressing the comments and for re-analysis of the data.